# Fifteen-, Ten-, or Five Minute City? Walkability to Services Assessment: Case of Dubai, UAE

**Maram Ali** [1], **Tarig Ali** [2,*], **Rahul Gawai** [2] and **Ahmed Elaksher** [3]

1    Master of Urban Planning Program, American University of Sharjah,
     Sharjah P.O. Box 26666, United Arab Emirates; g00095363@aus.edu
2    Department of Civil Engineering, American University of Sharjah,
     Sharjah P.O. Box 26666, United Arab Emirates; rgawai@aus.edu
3    Department of Engineering Technology and Survey Engineering, New Mexico State University,
     Las Cruces, NM 88003, USA; elaksher@nmsu.edu
*    Correspondence: atarig@aus.edu

**Abstract:** The rapid urbanization growth in Dubai has resulted in connectivity issues and is therefore considered tremendous development pressure. That has led the local authorities to set a vision for Dubai as a 15–20 min city by 2040. In a 15 min city, all services can be reached within 15 min of travel time using sustainable mobility means, including walking, cycling, or electric biking. This study aims to assess the current walkability situation within 15 min in the most significant parts of Dubai. The study considered 13 communities, including Bur-Dubai and Business Bay, which were selected to represent ungated communities and eleven major gated communities. Those neighborhoods were selected based on the developments' socio-economic status and population density. The assessment considered 14 essential services, grouped into five categories: educational, health, social, entertainment, and religious. The data for this study was collected through desktop research, site visits, and residents' interviews. The data layers were prepared in ArcGIS Pro 3.0, which was used to perform the network analysis. The results indicate that 28.25% of residents in the ungated neighborhoods have access to essential services within 15 min, similar to gated communities where residents rely on cars to access many services. Furthermore, results suggest that service distribution patterns and walkability infrastructure outside these communities should be developed to obtain higher walkability indicators.

**Keywords:** urban design; sustainability; sustainable development; gated community; walkability; Dubai; United Arab Emirates



## 1. Introduction

Sustainability is broadly defined as "meeting the needs of the present without compromising the ability of future generations to meet their own needs" [1]. However, it refers to three distinct areas: social, economic, and environmental, which are normally called the three pillars of sustainability [2]. Academic endeavors highlight the importance of ease of sustainable access to and within cities to alleviate environmental and social problems [3–6]. Walkability, which is the ability to walk comfortably to nearby services in a neighborhood, is a core urban design element because it is advantageous on three fronts: health, livability, and sustainability [7]. From environmental aspects, and according to the IPCC report for 2022 [8], the concept of walkable cities will play a pivotal role in shaping resilient cities by decreasing $CO_2$ emissions and reducing the factors contributing to climate change because the main source of these emissions is automobility. Implementing the idea of the 15-min city, urban sprawl, increasing population density, and considering key planning principles to achieve complexity remain the main challenges [9]. In other words, finding sustainable transportation solutions that are convenient for residents would help reduce emissions, which have adverse effects on the local climate [10]. There are evident side effects of

traditional transportation in Gulf cities such as Dubai, Abu Dhabi, and Doha, which have experienced excessive development pressure over the last 20 years [11]. By adopting the 15-min city concept in urban planning and infrastructure development, two of the United Nations' sustainable development goals (SDGs) would be met. These are goals "3": good health and well-being, and "11": sustainable cities and communities. For these reasons, the 15-min concept has become the most adopted concept in sustainable transportation and generally the most popular concept linked to sustainable cities.

The concept of a 15-min city has recently become popular. However, this notion has actually been inspired by earlier urban planning models, specifically Garden City by Ebenezer Howard [12], the neighborhood unit by Clarence Perry [13,14], which inspired the urban planning of many post-war British and American cities; the gravity theory by Walter Christaller [15], the urban diversity stated by Jane Jacobs [16], the geography of time by Torsten Hägerstrand; the rules for urban space and its indicator by Christopher Alexander and Jan Gehl [17], and the pedestrian pocket drafted by Peter Calthorpe [18], the symbiotic relationship between public transport and the spatial distribution of services in transit-oriented development [19], the principles of New Urbanism founded by Andrés Martin Duany and Elizabeth Plater-Zyberk [20], and Smart Growth [21]. Similar versions of 'urban cells' or 30- and 20-min neighborhoods have emerged globally in the past decade. Fifteen-minute cities are alternative forms of a more general concept, which refers to the essential service catchment areas where inhabitants can access primary amenities in a given period of time [22]. This research focuses only on walkability (cyclability and electric bikes are not considered). Most literature considers walking within a time frame of $\frac{3}{4}$ mile [23,24]. Of course, the accessible time differs between the various sustainable modes of transport as a sustainable travel mode to access essential services (health, education, religious, social, and entertainment). In this context, it is vital to highlight that the social services include transit stops, which are considered an indicator of neighborhood connectivity (transit stops could be bus or metro stations) [6], to allow the residents to access the services at the city level (universities, office zones, etc.).

This research aims to determine whether Dubai neighborhoods are walkable within 15 min, a key indicator of sustainability. The literature on Dubai does not address this urban planning and design aspect. Few authors highlighted a general overview of Dubai's walkability infrastructure without assessing and measuring the current walkability situation for Dubai [25–28]. The fastest development in Dubai makes it different from the long development process in other cities (American and European cities) [29]. The false-color map in Figure 1 shows Dubai's massive, dramatic, and quickest development in 1992, 2002, and 2022, respectively. In the earlier image, an empty desert fills the lower right part of the map as the cityscape hugs the coast. The final image shows the area almost served with transportation infrastructure, water development, and irrigated fields as vegetation spreads.

By incorporating health, education, entertainment, and social and religious needs into a 15-min city, self-sustaining communities can be developed. The only way to achieve this vision is by blending multiple types of urban zones and controlling services through multiple centers [16]. However, the plan for Dubai 2040 aims to apply Dubai's 15–20 min policy to all of Dubai's neighborhoods by developing sustainable means of transportation infrastructure and re-planning the locations of several essential services.

Dubai Master Plan 2040 involves creating integrated service neighborhoods with all the required facilities and increasing the population density around metro and bus transit stations. The master plan document defines the 20-min city as "a city where residents can travel to their destinations on foot or by bicycle in 20 min". The plan's vision is to promote 80% accessibility of their daily requirements within a maximum of 15 to 20 min. The plan expects to enhance the capacity of the service in urban areas and add new regulations and codes to get developing permits [30]. His Highness Sheikh Mohammed Bin Rashid Al Maktoum, the Ruler of Dubai, has approved phase II of the master plan, which will be

adopted in all of Dubai. It mainly focuses on the residential neighborhood sector, and the development agendas are based on the 15–20 min concept. The plan's agenda includes:

- Ensuring that resources are used in the best-planned manner possible.
- Promoting attractive, safe, and welcoming communities.
- Increasing the vegetation and the landscape areas to enhance the quality of the inhabitants' and tourists' lives.

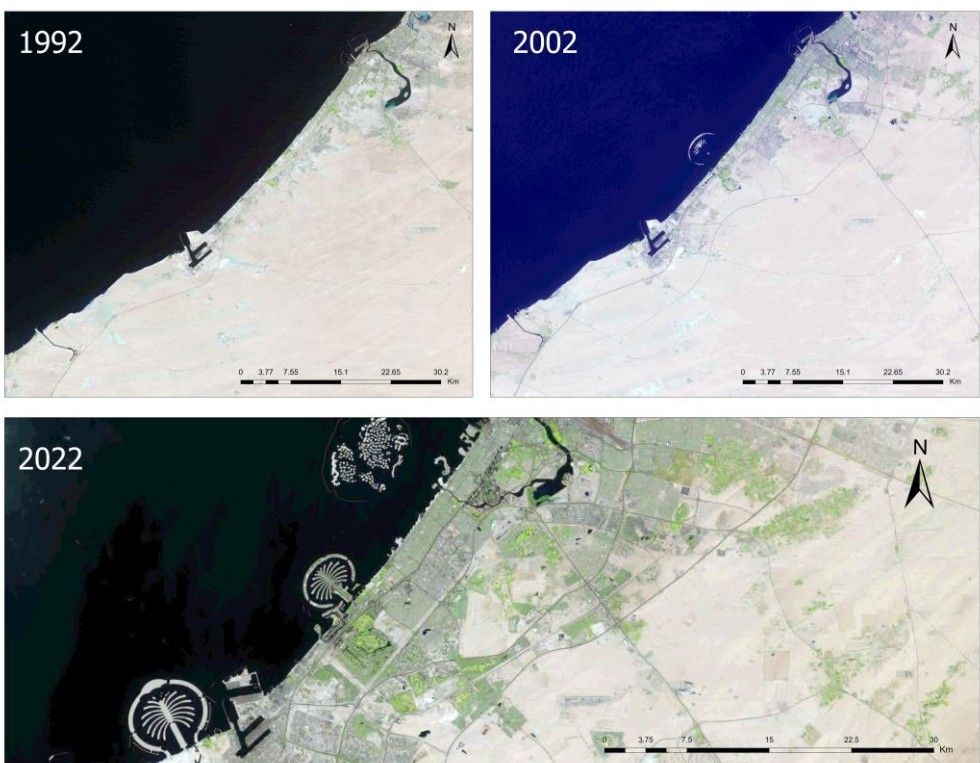

**Figure 1.** Dubai's rapid development over the past few decades, with significant changes observed in 1992, 2002, and 2022.

The master plan has the further goal of providing sustainable and flexible mobility to foster international investments and tourism activity in Dubai. In addition, this safeguards the Emirate's cultural and urban heritage by strengthening citizens' sentiments toward older neighborhoods and upcoming communities, developing comprehensive legislation, and planning a governance model to support sustainable development and growth. The current and upcoming challenge for the neighborhoods in Dubai is to ensure that all essential amenities and services are within a 15-min distance from the residents. Dubai's population is growing rapidly, and all of its neighborhoods are experiencing the fastest increase in population [31,32]. Figure 2 illustrates the annual growth rate of 1.58 to 1.48% in the last four years.

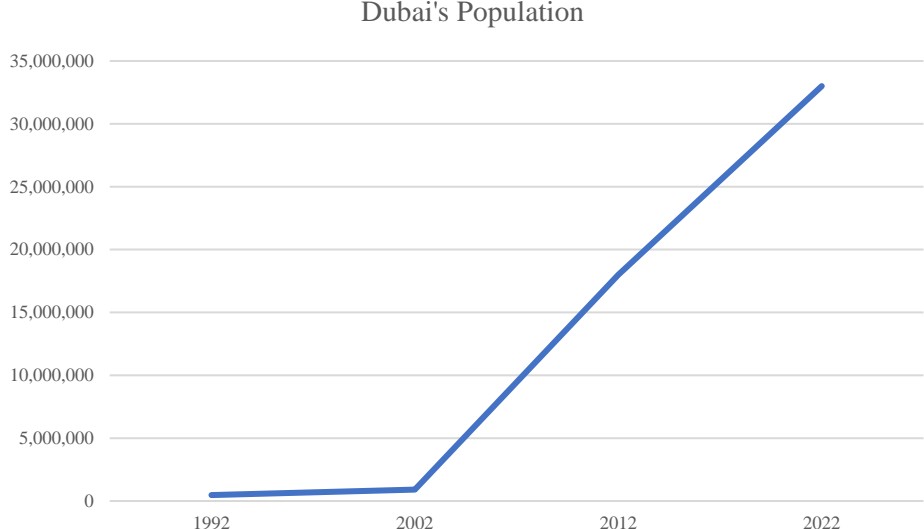

**Figure 2.** The population of Dubai has significantly increased over the last decade (from 1992 to 2022).

## 2. Methodology

The research seeks to assess the walkability of essential services in Dubai within 15 min, showing the proportion of residents who can access a certain service within a specific time frame. For that reason, this approach can be valuable during the analytical phase to evaluate the extent to which a city can be determined as a 5, 10, or 15-min city relative to the zonal allocation of specific services. The methodology's indicative implementation will be applied to some of Dubai's ungated and gated communities. The method will greatly impact the planning discussions while highlighting service distribution efficiency.

Dubai is located in the northern half of the United Arab Emirates, with a coastline that extends for about 70 km on the southwest coast of the Arabian Gulf. Figure 3 shows the geographical location of the city and the administration division for Dubai by neighborhoods. Dubai is also considered the country's most densely populated city, with a total area of 4114 square kilometers. In May 2023, Dubai's population stood at 3,578,354, and the population density was approximately 762.6 individuals per square Kilometer [31]. For several reasons, Dubai might be the prime candidate for a 15-min city case study. First, the high level of car-dependent infrastructure: "During the forecast period of 2022–2027, the UAE market is projected to grow at a Compound Annual Growth Rate (CAGR) of around 11%, from USD 20.03 billion in 2021 to USD 32.07 billion by 2027" [33]. Dubai is a city known for its heavy traffic, as it is predominantly automobile-oriented. However, in the early 21st century, new transportation networks are fully automated, and high-tech metro rail controlled by sensors, cameras, radars, and artificial intelligence systems has helped to alleviate some of the frustration of getting around certain parts of the city. Secondly, the variety of urban fabric and planning concepts differed from one area to another. Dubai consists of two fundamental types of neighborhoods:

(a)　Ungated communities: urban and social housing units planned for low, medium, and high-density populations Those areas could be considered for integrating residential, commercial, recreational, and other communities. These neighborhoods could be categorized into two types: the first one existed and mostly developed between 1970 and 2000 (Bur Dubai is one of those neighborhoods), which is shown in Figure 4 [34]. The second is that the development's infrastructure was completed between 2003 and 2008 (Business Bay and Downtown Dubai). Still, they boomed between 2015 and 2018, and until now, most of their parts are under development, and construction in those areas is progressing gradually. This research considers both categories by choosing Bur Dubai and Business Bay (Figure 5). Business Bay is a rapidly developing central business district in Dubai, encompassing a vast area of 4,360,000 m$^2$ with

over 240 buildings comprising both commercial and residential developments. The district is located immediately south of Downtown Dubai and is built around the extended and dredged Dubai Creek. The total gross leasable area is 7,290,000 m$^2$, and the projected development population is over 191,000, with an estimated population of employers and others reaching 110,000, making the total population more than 300,000. Bur Dubai is another important district located on the western side of Dubai Creek. It is a historic area that translates to Mainland Dubai, referring to the traditional separation of Bur Dubai from Deira by the Dubai Creek.

(b) Gated communities: These communities (Figure 6) have almost similar socioeconomic levels. It consists of private villas with luxurious services (such as Damac Hills) with a low-density population [35], or it could be a combination of private villas and medium-density social housing (Al Shorouq and Al Ghroobcases). Those developments were constructed in a time frame of 4 to 7 years. The services in this type of community do not include the basic needs (social, health, educational, and religious services), but they do include entertainment services such as sports facilities, leisure parks, spas, and cafes. This study considers eleven gated communities (Emirates Hills, DAMAC Hills, Arabian Ranches, Dubai Hills, The Springs, Akoya Oxygen, Al Barari, Meydan, Mudon, Jumeirah Golf, and The Meadows) and assesses their walkability indicators.

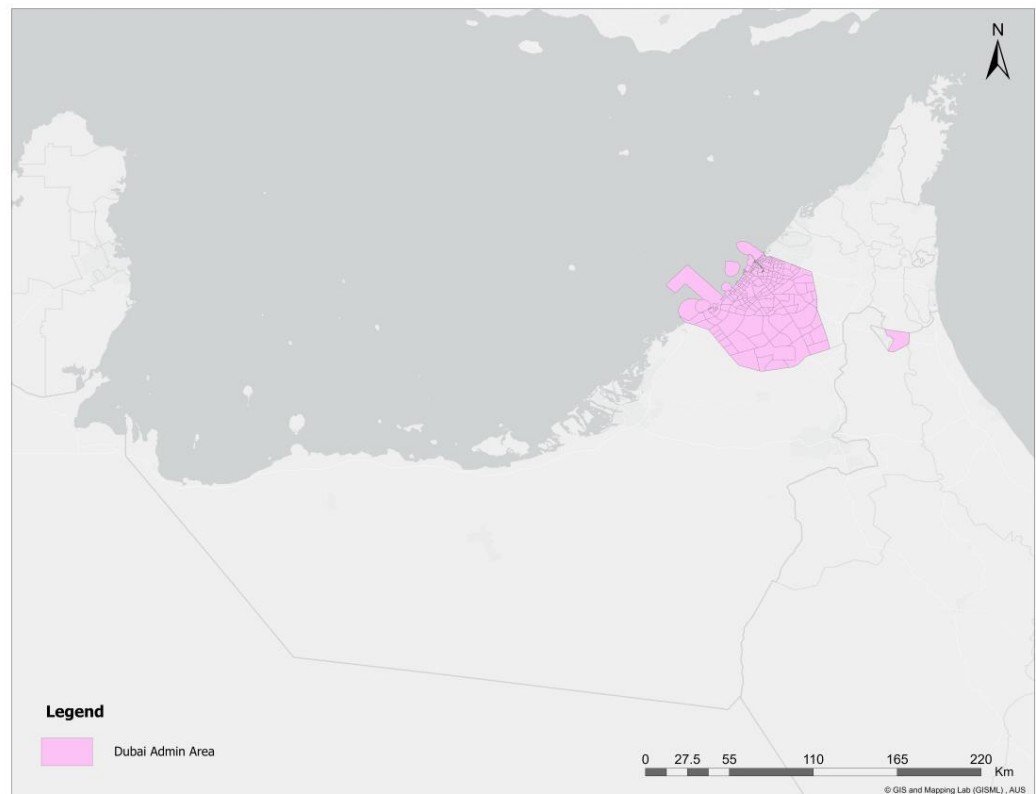

**Figure 3.** The precise geographic location of Dubai (Dubai's boundary is discontinuous because of the isolated location of Hatta).

Since the two types of Dubai's developments are entirely different in their urban structure and service infrastructure, the measuring methods for their potential to be 15-min walkable communities should be different too. It varies from using GIS techniques (where the neighborhoods are mixed-used and the network analysis could be applied), which is applicable in the ungated neighborhoods, to measuring the walkability indicators by using a walkability assessment framework (where the network analysis could not be applied), which is the case for gated communities.

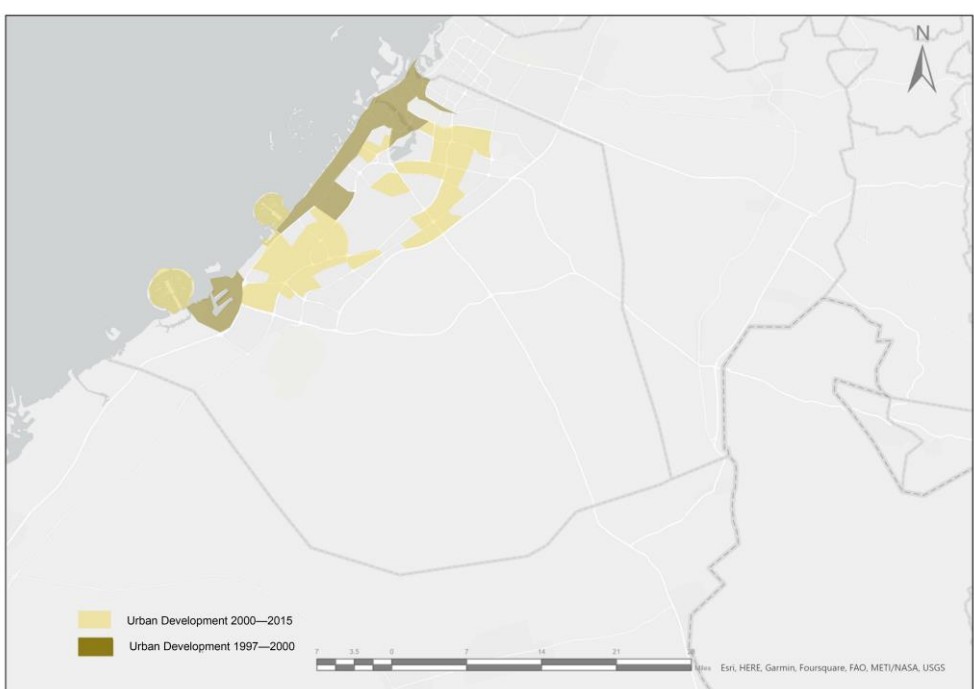

**Figure 4.** Dubai experienced significant development from 1970 to 2000 in a particular region.

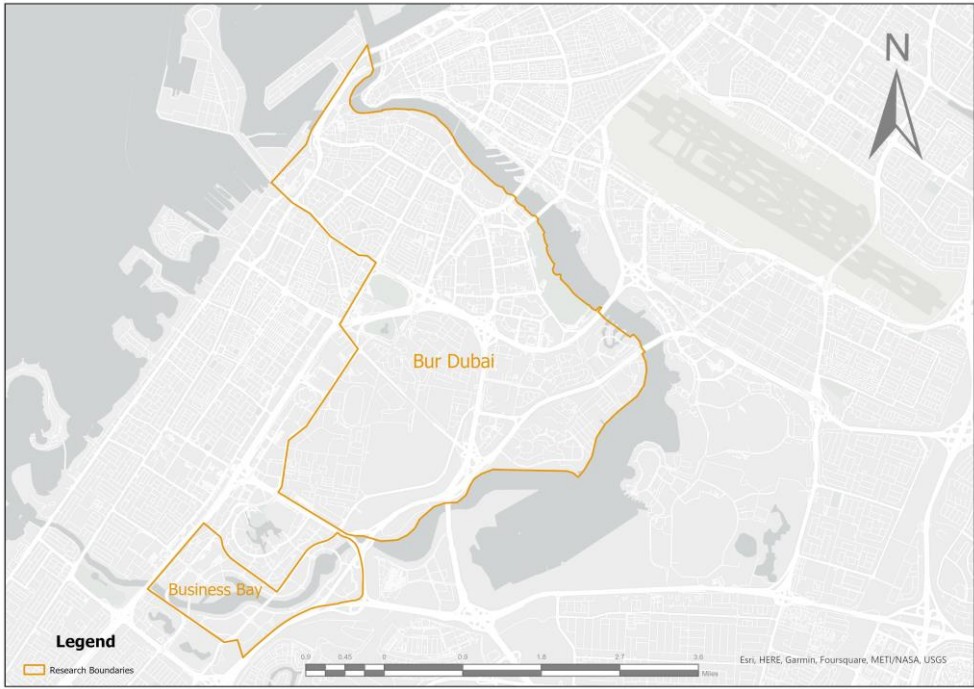

**Figure 5.** Bur Dubai and Bussiness Bay boundaries.

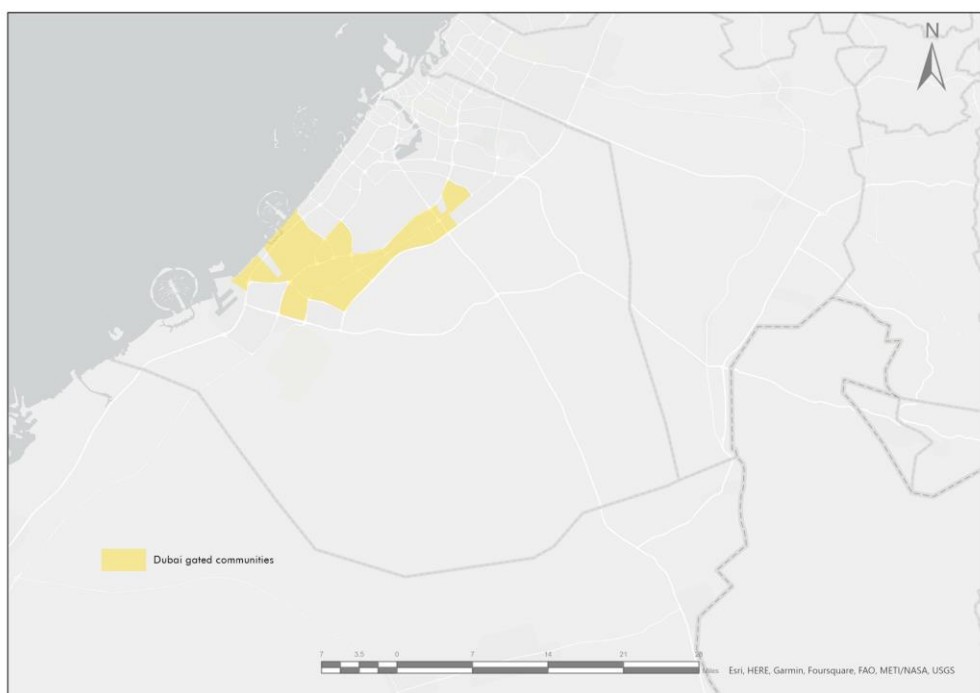

**Figure 6.** The geographical sprawl of Dubai's gated communities in 2023.

## 3. Analysis and Results

### 3.1. Ungated Community (Neighborhoods) Analysis

This study evaluates the relationship between mixed land use and physical accessibility by assessing the walkability indicators in Bur Dubai and Business Bay. Map layers have been used to document the evolution of urban developments. As appropriate, all of the map layers were created by ArcGIS Pro 3.0, using some confidential sources of data in addition to the available data posted on the Dubai Municipality website [36,37]. Besides ArcGIS Pro 3.0, other digital tools like Google Earth have been used to create some of the layers used in this study. Bur Dubai and Business Bay neighborhoods were selected in this walkability study to represent ungated communities for the following reasons: (a) high population density; (b) the sprawl of the residential areas with limited planning considerations (in the Bur Dubai case) for the city compactness to encourage active mobility methods instead of automobility; and (c) the central location of the two neighborhoods.

The following tasks were carried out in preparation for the analysis:

a.  Identifying Bur Dubai and Business Bay boundaries. Broad areas in Bur Dubai are hosting no residents. Those areas identified do not affect the analysis results (Figure 7).

b.  Identifying the locations of services for which accessibility is to be calculated. Fourteen types of services have been considered, which belong to the following main categories:

■    Education services: daycare centers, primary, and secondary schools.
■    Health services: health centers, hospitals, and pharmacies.
■    Social services: groceries, daily needs shops, bus transit stops, post office, and police stations.
■    Entertainment services: parks, cinemas, sports facilities (swimming pools, gyms, football courts, basketball courts, etc.)
■    Religious services: mosques and churches.

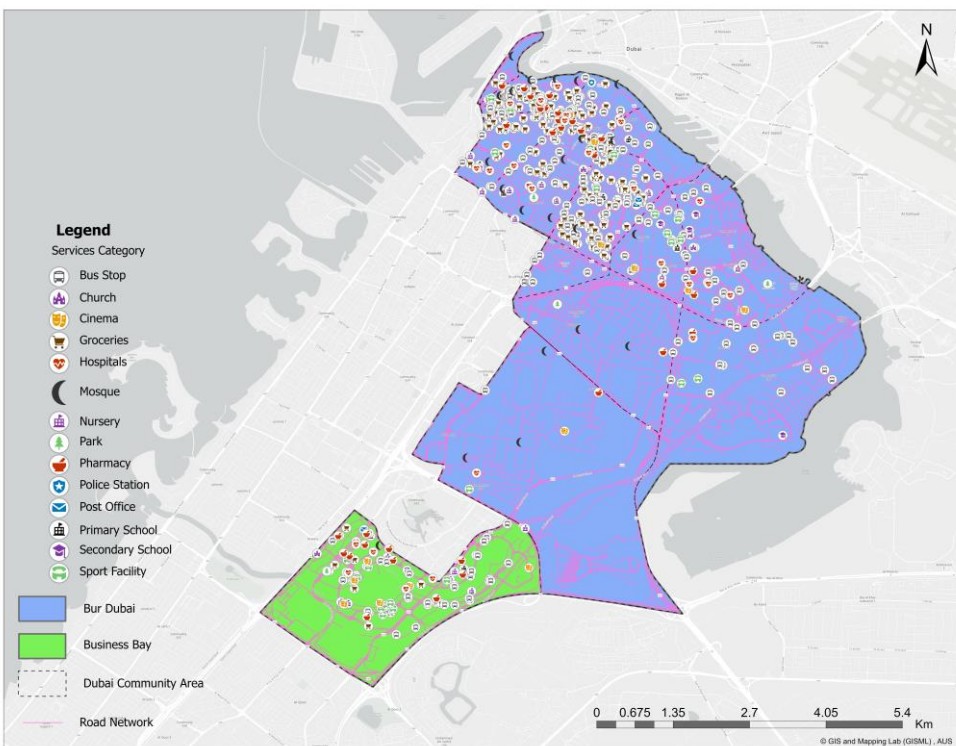

**Figure 7.** The geographical distribution of the services in Bur Dubai and Business Bay.

These services were selected after a comprehensive study through the 15-min neighborhood literature, specifically [7,16,17,38–43]. The 14 prospective services presented for the 15-min concept are based on the following two major points of reference:

(a) Accessibility at a neighborhood level, which excludes city-wide services like universities and malls.

(b) Georeferenced data, which determines the spatial requirements, existed within the neighborhoods; therefore, services such as libraries and social consultation centers were excluded due to their unavailability in the study area. Regarding commercial services, the only ones considered for daily needs are supermarkets (widely used at the neighborhood level in Dubai) and some types of commercial services, such as open markets. While other commercial shops are not considered in the analysis, such as car maintenance and clothing shops, they are not essential services due to their irregular needs.

Furthermore, food and beverage services were excluded from the analysis due to positive reachability results in all neighborhoods that cover 90% of the residents within a 5-min time frame. Table 1 highlights the density of service allocation between the Business Bay and Bur Dubai communities. The number of essential services differs among the targeted study areas. For example, the number of grocery stores in Business Bay is 14, while in Bur Dubai it is 143. Moreover, this is a common occurrence across all 14 services examined.

**Table 1.** The count of the five destination categories in Bur Dubai and Business Bay.

| Service Category | Service Name | Bur Dubai | Business Bay |
|---|---|---|---|
| Education Services | Nurseries | 17 | 9 |
| | Primary Schools | 8 | 1 |
| | Secondary Schools | 7 | 1 |
| Religious Services | Mosques | 24 | 1 |
| | Churches | 6 | 1 |
| Health Services | Hospitals and Health Centres | 22 | 6 |
| | Pharmacies | 23 | 11 |
| Social Services | Police Station | 2 | 0 |
| | Post office | 2 | 2 |
| | Bus Stops | 161 | 37 |
| | Groceries | 143 | 14 |
| Entertainment services | Cinemas | 8 | 6 |
| | Sport Facilities | 19 | 10 |
| | Parks | 4 | 2 |

As shown in Table 1, the locations of the five service categories vary significantly. Every service was digitized as a point matching the entrance's geographical reference point (Figure 7). As shown in the figure, Dubai is divided into districts (community boundaries) to be more accurate, considering the spatial distribution of the services, and the boundaries were created according to Dubai's Municipality [44].

Determining the 5-, 10-, and 15-min walking time from residential areas to the services using the network analysis was done as follows:

1. Identify the relevant walking radius calculated among the road network that serves all the Bur Dubai and Business Bay blocks and zones (both residential and non-residential) having a walking distance of 0.40 km, 0.80 km, and 1.20 km (taking into account the standard walking speed of 4.8 km/h) that correspond to 5, 10, and 15 min, respectively (14 destination types of services are considered).

2. Create a network dataset in ArcGIS by taking the spatial reference for the services from Google Maps and double-checking whether the service works by contacting the service and sometimes through site visits.

3. To optimize the accuracy of the walkability assessment, the centroid point for each district was used, and then the network analysis was done for each service separately within every district (Figure 8).

4. Then, incorporate the information in ArcGIS Pro 3.0 to georeference all the facilities as point layers within the Bur Dubai and Business Bay districts. Figure 9 shows the road network and residential buildings in the districts of Bur Dubai and Business Bay. The residential buildings were excluded and georeferenced as polylines to keep their existing dimension. The buildings are also digitized to measure the walkability distance from their spatial dimension as a starting point for the resident journey.

5. Utilized the network analyst in ArcGIS to calculate service accessibility through the roads. The entrance point is considered only for the neighborhood parks and hospitals because the large polygon area may interpret the analysis results accurately. Although Dubai's master plan aims to achieve a 20 min city concept, many case studies proved that the 20 min city has failed to consider mobility diversity and is not reachable for senior citizens and children [20]; the research used cut-off times of 5, 10, and 15 min only (Figure 10). Furthermore, the analysis identified walkable routes within 0–15 min from services to residential buildings in the neighborhood (Figure 11).

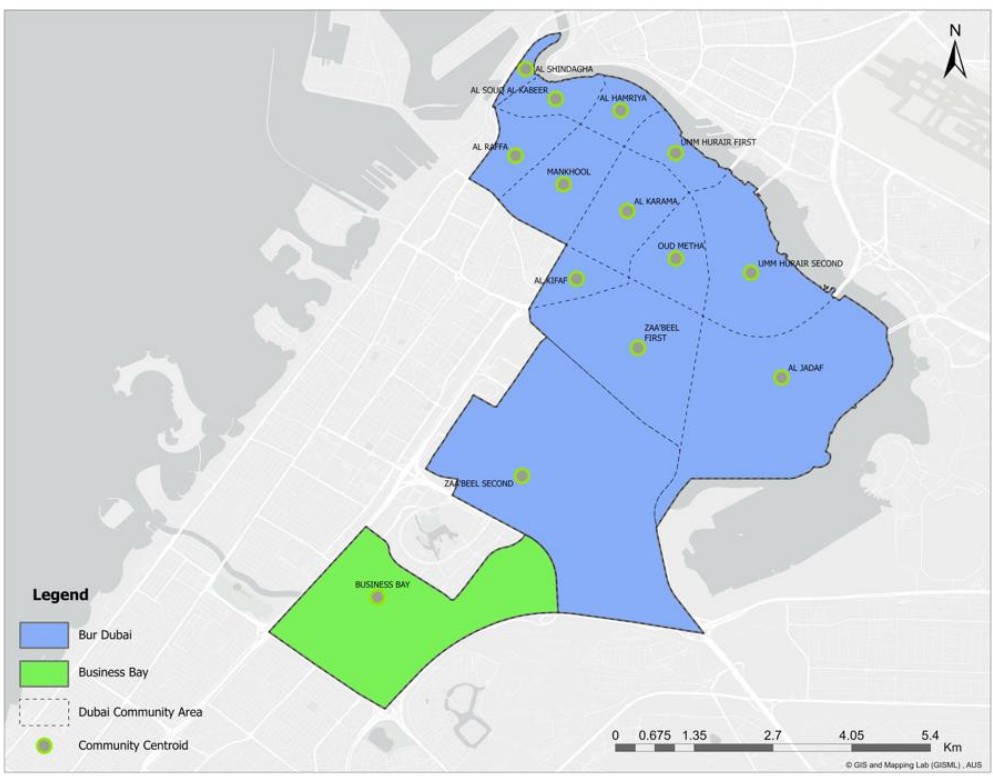

**Figure 8.** The boundary between the Bur Dubai and Business Bay districts, as well as their centroid point.

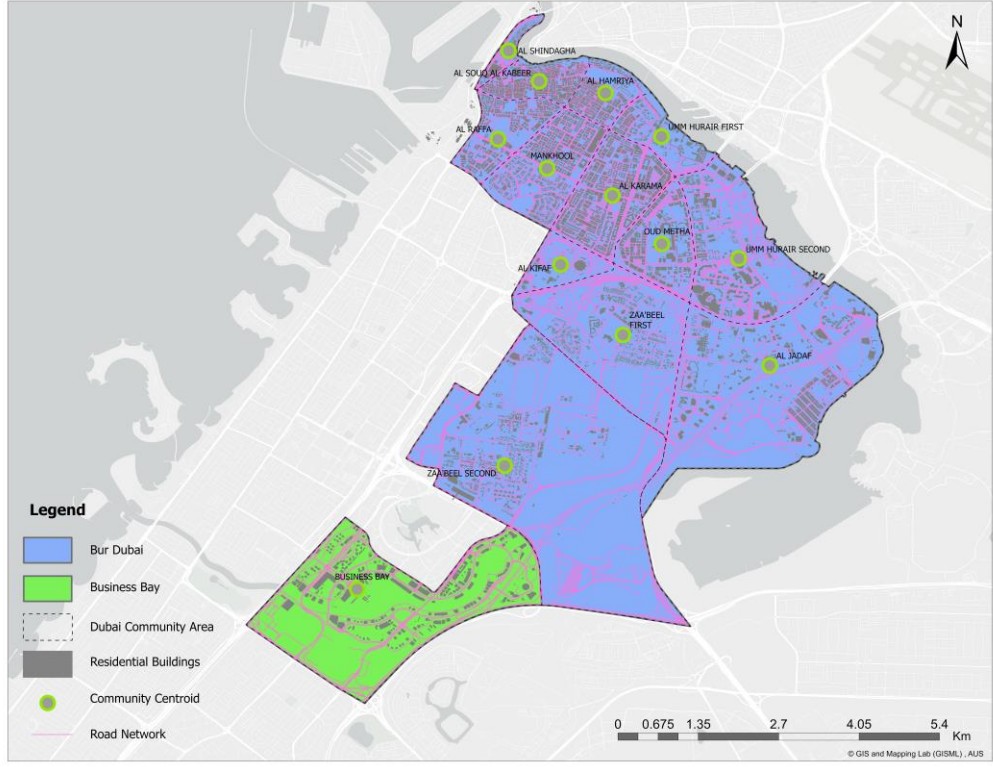

**Figure 9.** Road and residential buildings in the research area.

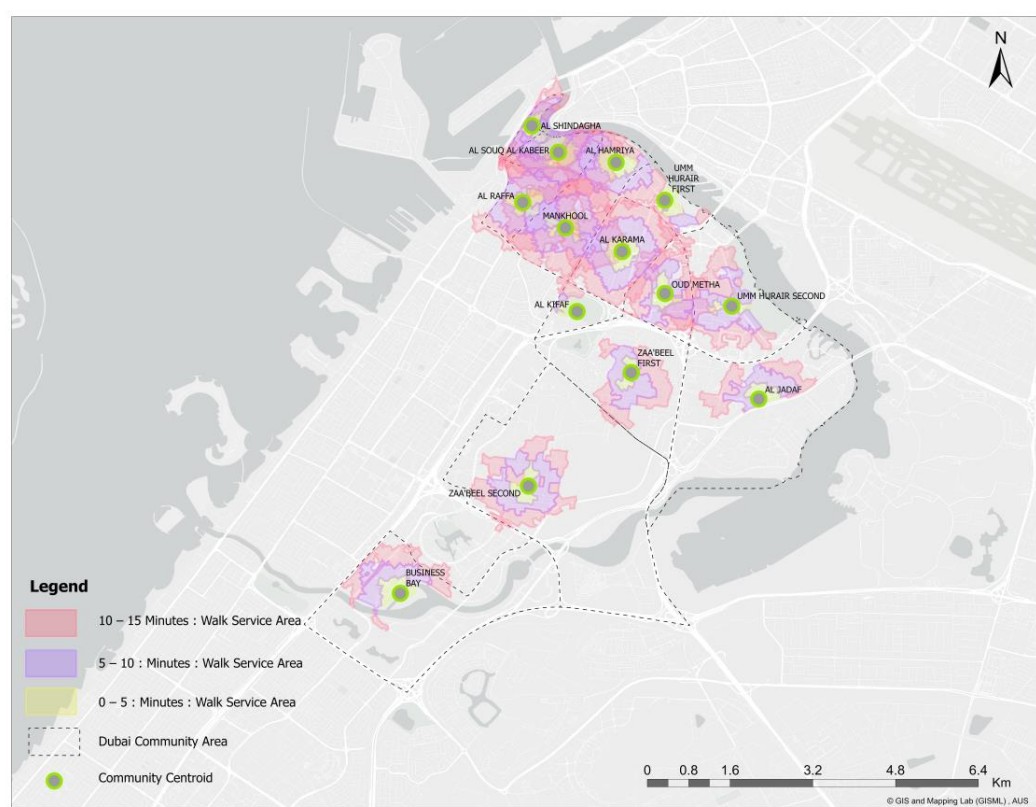

**Figure 10.** The network analysis for the services using the time frame of 0–15 min in Bur Dubai and Business Bay.

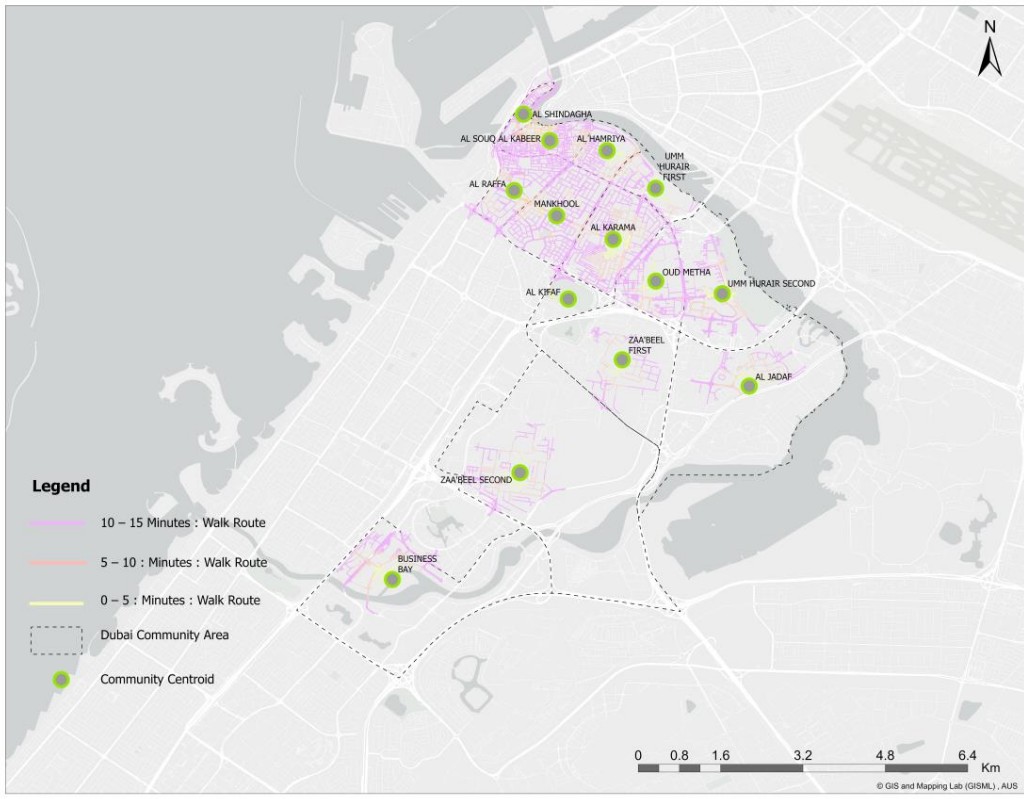

**Figure 11.** The walking routes to the services in Bur Dubai and Business Bay.

### 3.1.1. Results of the Ungated Communities Network Analysis

This section aims to conclude the findings of the ungated communities' network analysis by focusing mainly on the population percentages, specifically those served within the 5-, 10-, and 15-min time frames. First, the study results show that almost all of the service numbers could be accessible within a 15-min time frame (Figure 12). The entertainment services have shown higher percentages of service locations with less time accessibility. In contrast, social services showed that more than 40% of the locations of the services are accessible within the 15-min time frame. Furthermore, results showed that the lowest service accessibility rates are for secondary schools, parks, and cinemas. Second, according to the estimated population density by [44], 28.25% of the population in Bur Dubai and Business Bay can access all of the services within 15 min of walking (143,017 out of 324,520). At the same time, 71.75% of residents would need to walk for more than 15 min to access some of the services (Figure 13). As illustrated in Figure 14, the services in the area can be accessed within a 10–15 min walk, which is the highest time frame required. However, there is a lack of accessibility to certain services such as churches, secondary schools, parks, and police stations within 0–5 min of walking distance.

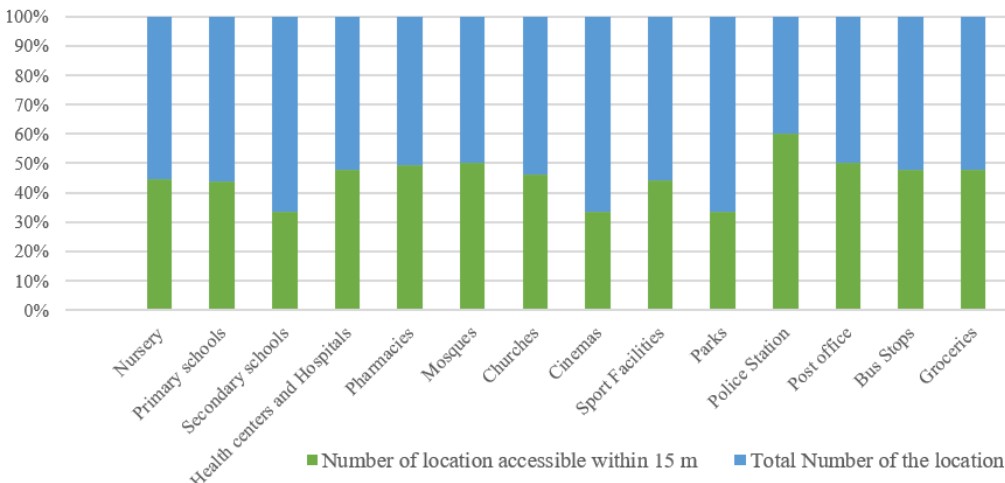

**Figure 12.** The percentage of trips (all modes) per destination type.

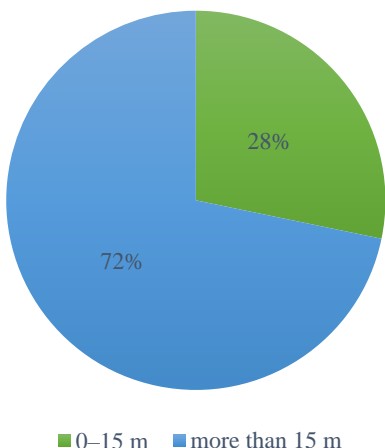

**Figure 13.** The percentage of population that is able to access essential services within a certain period of time.

Third, to calculate population walkability percentages within the time frames (0–5), (0–10), and (0–15), Bur Dubai and Business Bay were divided into districts according to Dubai's Municipality [44]. To optimize the accuracy of the walkability assessment, the

centroid point for each district was used, and then the network analysis was done for each service separately within every district.

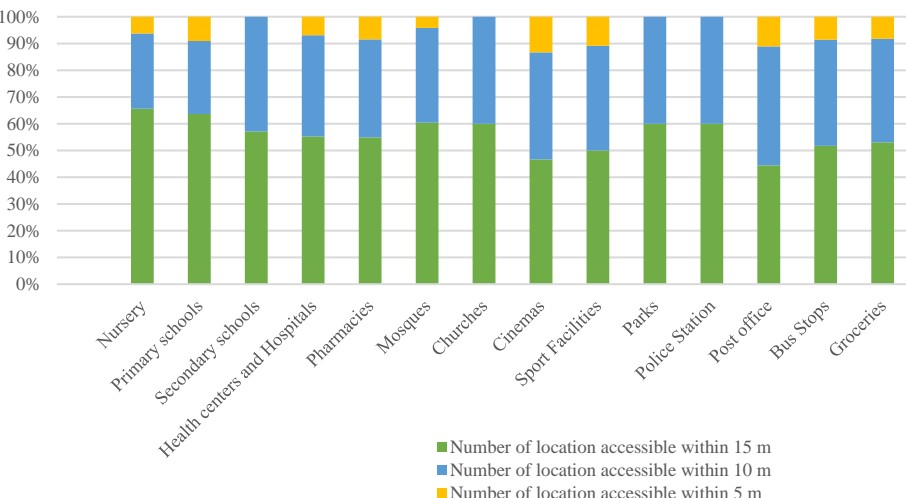

**Figure 14.** The percentage of the population with access to a service within a given walking time varies.

Fourth, the percentage of residents who can access specific services was calculated for each facility's service area (Table 2). Noticeably, the percentage differs significantly among the services within 5 min, which can be summarized as follows: (A) Primary schools have the highest service accessibility, with 21.74% of the population being able to access it. In contrast, secondary schools, churches, parks, and police stations are not accessible within that time limit. (B) It is clear that the number of accessible, specific services is not always related to the percentage of the population with accessibility. For example, five sports facilities could be accessed within 5 min of walking, but the population that can access them is only 9.18%. While only one primary school is accessible within the same time limit, 21.74% of the inhabitants have this accessibility.

Then, for the service accessibility within the 0–10 min time limit: (A) The primary schools are still the ones with the highest service accessibility rate, with 39.22% of the residents able to access it. (B) The number of bus stops and groceries in this network is high: 130 and 160 services, with (18.81%) and (29.56%) for each service, respectively. In this context, it should be highlighted that the health services accessibility has almost doubled from 0–5 and 0–10 min walking, from 8.44% to 16.67% in the health centers and hospitals case, and from 14.46% to 24.75% in the case of pharmacies. (C) The churches have the lowest accessibility rate, with only 2.10% of the inhabitants accessing them within (0–10) minutes, while the parks and the police stations have the lowest amounts of locations in this time frame of walkability, with only two locations for each facility.

Last, the accessibility within a (0–15) minute time frame: (A) The accessibility of the park increased dramatically from 3.04% in the (0–10) m to 29.1% in the (0–15) m walking, although only three parks are located within the boundaries of the (0–15) minute network analysis. (B) The highest service accessibility is still the primary schools, with 51.03% of the inhabitants walking to reach them as their destination. (C) The churches have the lowest service accessibility, with a 3.50% walkability rate. (D) The rest of the service's accessibility percentages fluctuate between 18.86% for the nurseries and 44.07% for the groceries.

In summary, the significant difference in the walkability rates of various services is the primary reason behind the reduced average residents' walkability percentage. The overall percentage for a walking distance of 0–5 min is just 7.14%, which is quite low. However, it increased by more than double to 17.47% for the 0–10 min. Finally, the walkability rate for a walking distance of 0–15 min is 28.25%. Although the number of services has doubled, the percentage increase is only slight with respect to the number of services.

**Table 2.** The percentages of the population who have access to each service within three time frames (0–5 min), (0–10 min), and (0–15 min).

| Accessible Location | 0–5 m | | | 0–10 m | | | 0–15 m | | |
|---|---|---|---|---|---|---|---|---|---|
| | Number of Location | Population with Accessibility on Average | Percentage of Population's Accessibility | Number of Location | Population with Accessibility on Average | Percentage of the Population's Accessibility | Number of Location | Population with Accessibility on Average | Percentage of Population's Accessibility |
| Education Services | | | | | | | | | |
| Nursery | 2 | 9858 | 3.04% | 9 | 33,657 | 10.37% | 21 | 61,191 | 18.86% |
| Primary schools | 1 | 70,558 | 21.74% | 3 | 127,289 | 39.22% | 7 | 165,602 | 51.03% |
| Secondary schools | 0 | 0 | 0 | 3 | 42,359 | 13.05% | 4 | 77,533 | 23.89% |
| Health services | | | | | | | | | |
| Health centers and Hospitals | 4 | 27,387 | 8.44% | 22 | 47,211 | 16.67% | 32 | 78,920 | 24.32% |
| Pharmacies | 6 | 46,929 | 14.46% | 26 | 80,308 | 24.75% | 39 | 108,531 | 33.44% |
| Religious Services | | | | | | | | | |
| Mosques | 2 | 23,466 | 7.23% | 17 | 47,211 | 14.55% | 29 | 69,409 | 21.39% |
| Churches | 0 | 0 | 0 | 4 | 6808 | 2.10% | 6 | 11,347 | 3.50% |
| Entertainment Services | | | | | | | | | |
| Cinemas | 2 | 19,715 | 6.08% | 6 | 55,567 | 17.12% | 7 | 82,774 | 25.51% |
| Sport Facilities | 5 | 29,796 | 9.18% | 18 | 66,257 | 20.42% | 23 | 103,625 | 31.93% |
| Parks | 0 | 0 | 0 | 2 | 9858 | 3.04% | 3 | 0 | 29.1% |
| Social Services | | | | | | | | | |
| Police Station | 0 | 0 | 0 | 2 | 40,175 | 12.38% | 3 | 78,099 | 24.07% |
| Post office | 1 | 19,715 | 6.08% | 4 | 73,325 | 22.60% | 4 | 118,462 | 36.50% |
| Bus Stops | 30 | 29,615 | 9.13% | 130 | 61,038 | 18.81% | 181 | 90,411 | 27.86% |
| Groceries | 34 | 47,912 | 14.76% | 160 | 95,941 | 29.56% | 219 | 143,017 | 44.07% |
| Population percentage with accessibility mean value | | 7.15% | | | 17.47% | | | 28.25% | |

### 3.1.2. Available Services and the Population

One of the most critical questions is: What is the relationship between the availability of the services and the number of residents targeted by the services? Figure 15 illustrates a negative trend in many services in all time frames considered for walkability. Sports facilities, primary schools, and post offices are always showing a negative trend within all the time frames considered, while bus stops, cinemas, parks, pharmacies, police stations, groceries, and hospitals are showing positive trends. In other words, when the number of services is increased, the number of residents who can access the services increases accordingly. The reason behind that is that many services were planned to be shared with various neighborhoods (cinemas, secondary schools, parks, and churches), so they were located outside of the highest-density population area. At the same time, other services (groceries and bus stops) were planned to be more concentrated in the high-density areas. In comparison, the post offices and the police stations are only located in the highest-density zone. High variations between the service counts and the resident's walking rate can be pointed out in Figure 15. There is a significant difference between the decrease in primary school attendance and the increase in grocery shopping. It can be concluded that adding more services is not essential for improving walkability, as the critical factor is the spatial location.

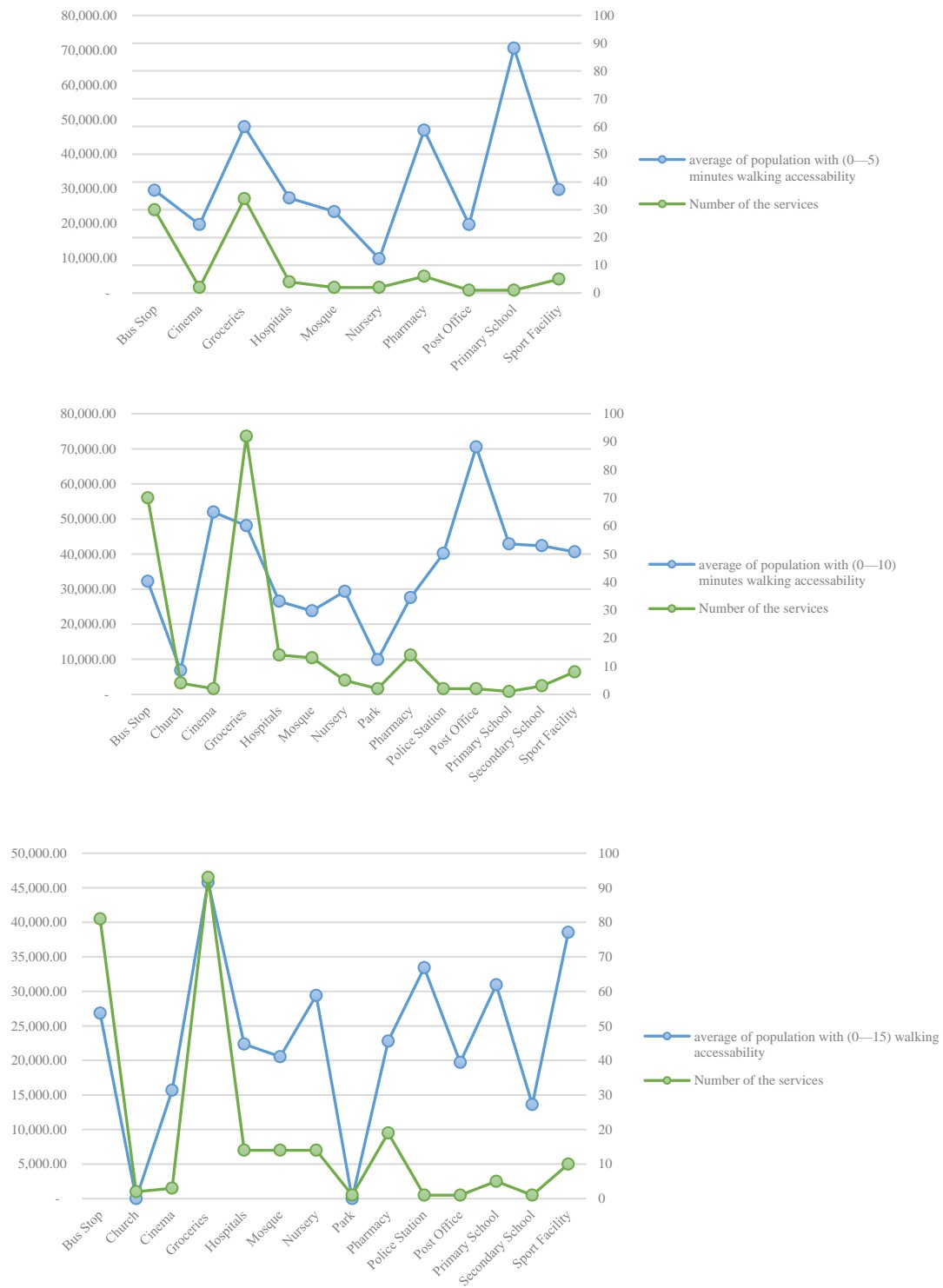

**Figure 15.** The relationship between population and accessibility.

### 3.2. The Gated Community Analysis

Across Dubai, the upper-middle class is exacerbating residential segregation and social divisions, creating new forms of segregation inside gated neighborhoods. Walled and fortified communities are by no means new in urban planning history. The evolution of gated communities began in cities through the Middle Ages and often featured walls for protection from intergroup hostility. Then, the concept was adopted in the US to secure properties and contain retirees' entertainment amenities [45]. In the UAE, the early

appearance of gated communities was in the 2000s [46], with 84% of the residents choosing to live there (unrelated to work or external hosting reasons). Their choices depended on many factors, such as cultural origin and socio-economic status [47]. These urban developments in Dubai now target a much broader and higher demand in the market in the name of sustainable communities. The gates were used in those communities to secure the residents with guards and with specially offered services with exclusive access for the residents of those neighborhoods. This would not promote accessibility to the open areas and may increase cultural and social intolerances [48].

The importance of considering gated communities in this research is that they account for more than 15% of the area of the residential neighborhoods in Dubai. This research considers eleven significant gated communities in Dubai: Emirates Hills, DAMAC Hills, Arabian Ranches, Dubai Hills, The Springs, Akoya Oxygen, Al Barari, Meydan, Mudon, Jumeirah Golf, and The Meadows (Figure 16). According to [49], the Arabian Ranches are the biggest ones in area, population density, and service range in the United Arab Emirates.

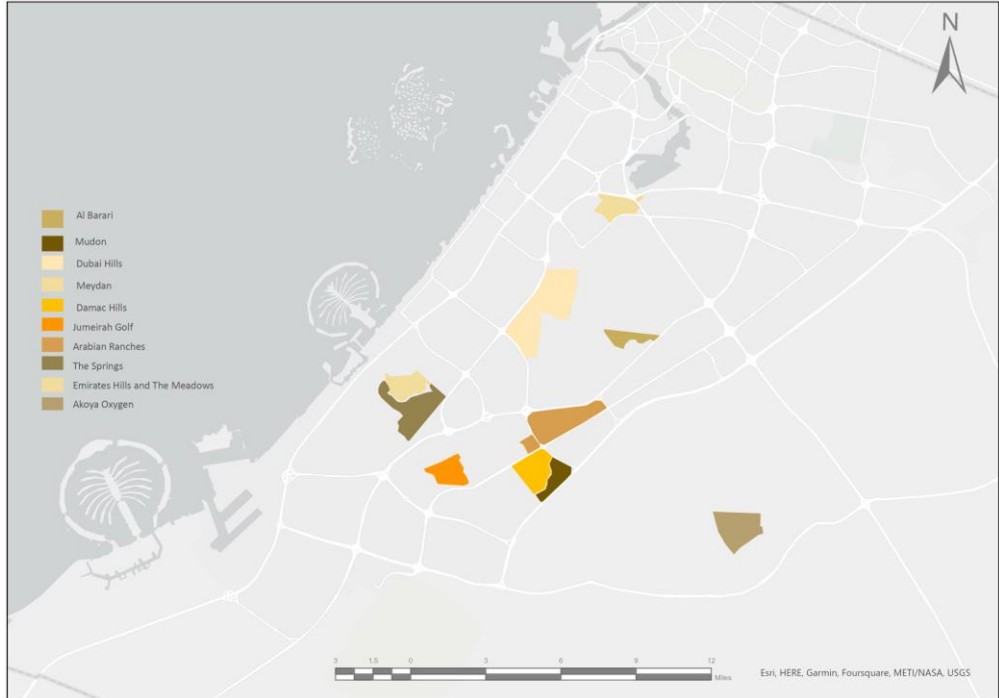

**Figure 16.** Research area (Dubai's gated communities, which are considered in the research).

Although those communities included considerable walkability infrastructure for their residents (safety, walkways, green spaces, lighting features, etc.), the offered services are mainly recreational and prestigious, as shown in Table 3. Arabian Ranches and the Springs are the only two gated communities that include healthcare and education facilities as part of their land use planning. Damac Hills, Dubai Hills, Akoya Oxygen, and the Meadows have considered only the education facilities as part of the neighborhood planning. It is clear that the consideration of basic needs is low; 18% considered health facilities, 54% regarded education facilities, and any clear consideration for social (even 45% have considered the groceries and daily needs shops, are not considered the walking distance and the service coverage area; Arabian Ranches are excluded) and religious facilities (100% have mosques, but none consist of churches). In contrast with the entertainment facilities, 100% of the gated communities are considered.

**Table 3.** The services provided in various gated communities.

| Gated Comunity | School | Nursery | Hospital | Health Center | Kids Play Area | Spa | Golf Course | Pools | Other Sports | Cafes | Retails | Post Office | Mosq Ues | Shopping Center |
|---|---|---|---|---|---|---|---|---|---|---|---|---|---|---|
| Damac Hills | | | | | | √ | √ | √ | √ | √ | | | √ | |
| Emirates Hills | √ | √ | | | √ | √ | √ | √ | √ | √ | √ | √ | √ | √ |
| Arabian Ranches | √ | √ | √ | √ | √ | √ | √ | √ | √ | √ | √ | √ | √ | √ |
| Dubai Hills | √ | √ | | | √ | √ | √ | √ | √ | √ | √ | √ | √ | √ |
| The springs | √ | √ | √ | √ | √ | √ | √ | √ | √ | √ | √ | √ | √ | √ |
| Akoya Oxygen | | √ | | | √ | √ | √ | √ | √ | √ | √ | | √ | |
| Al Barari | | | | | √ | | | √ | √ | | | | √ | |
| Meydan | | | | | | | | √ | √ | √ | | | √ | |
| Mudon | | | √ | √ | √ | | √ | √ | √ | √ | √ | √ | √ | |
| Jumeirah Golf | | | | | √ | | √ | √ | √ | | | | √ | |
| The Meadows | √ | √ | | √ | √ | | | √ | √ | √ | √ | √ | √ | √ |

In addition to the services in Table 3, it is essential to highlight the bus stops in the gated communities in Figure 17. It is evident that some bus stops are shared services between more than one community (the case of the springs with the meadows in Figure 18 and Meydan with Nad Al Shiba), which is only reachable for some residents in the 15-min walking time frame (Figures 18–20). In Arabian Ranches', the highest percentage of residents (90%) can access the bus stop service within 15 min (Figure 19). Jumeirah Golf, Mudon, Akoya Oxygen, Dubai Hills, and Emarites Hills are outside the bus service area (Figure 17).

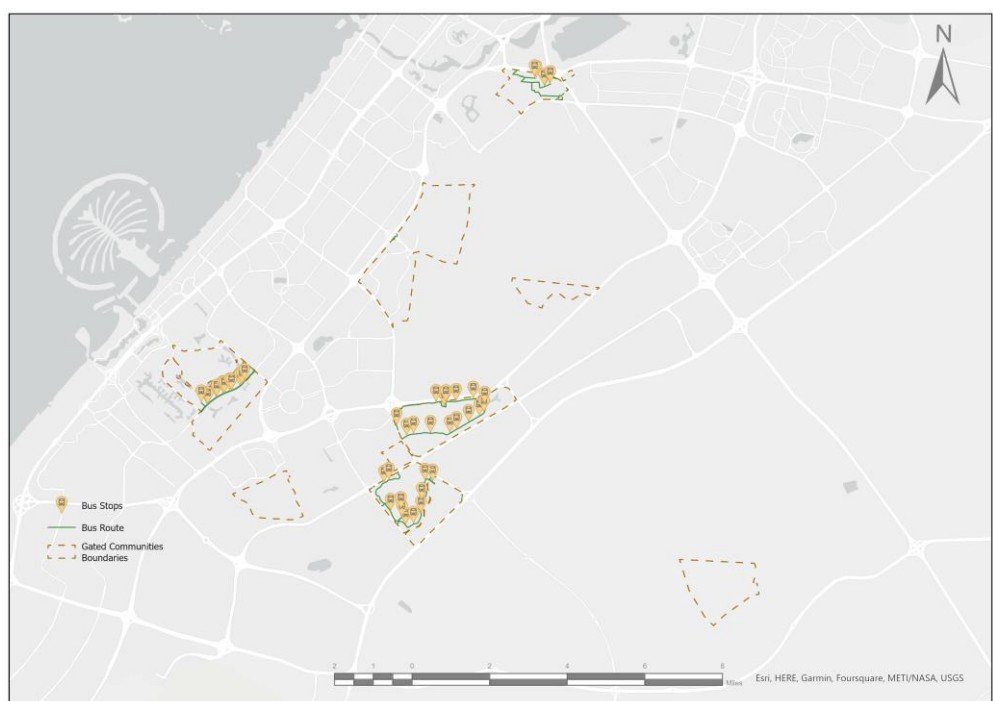

**Figure 17.** The provided bus stops in various gated communities.

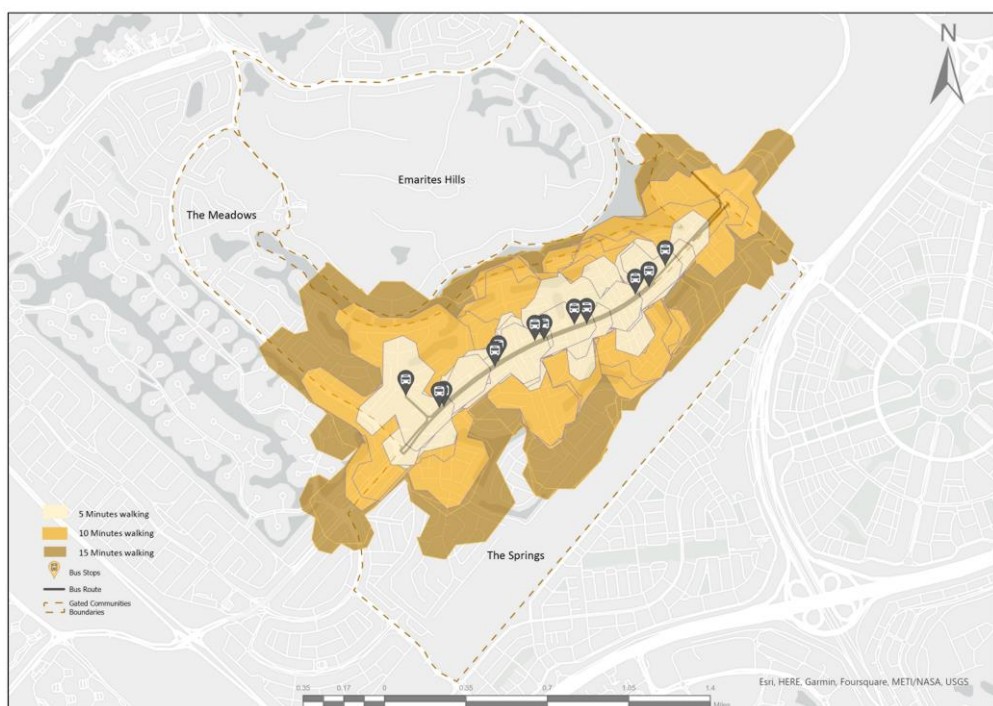

**Figure 18.** The bus stops walkability within 5, 10, and 15 min for the Springs and the Meadows.

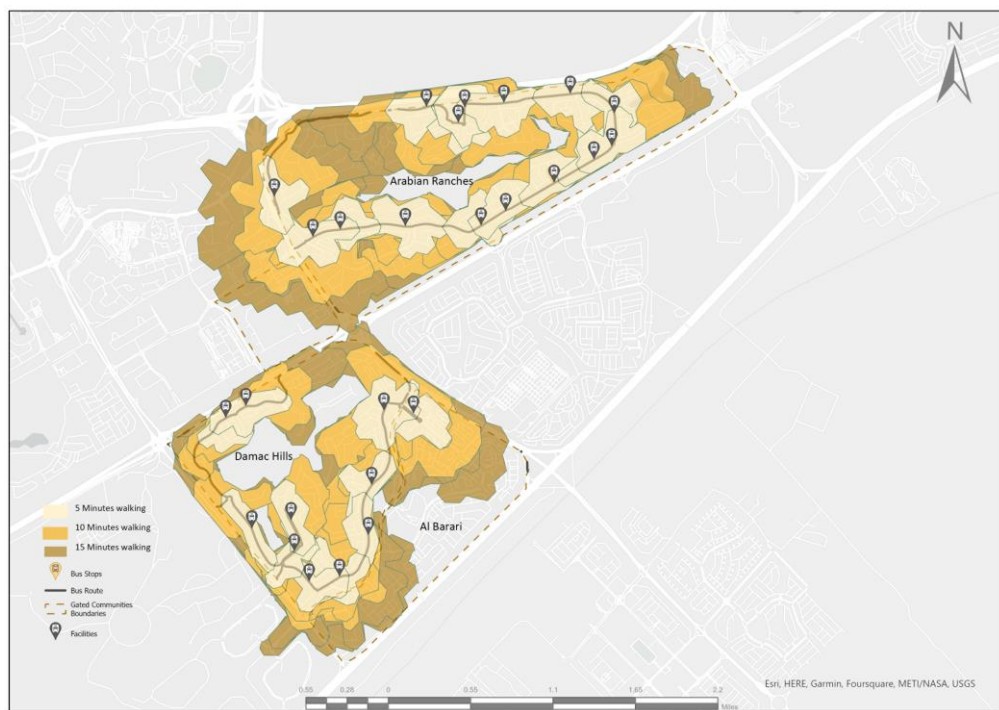

**Figure 19.** The bus stops walkability within 5, 10, and 15 min for Al Barari, Arabian Ranches, and Damac Hills.

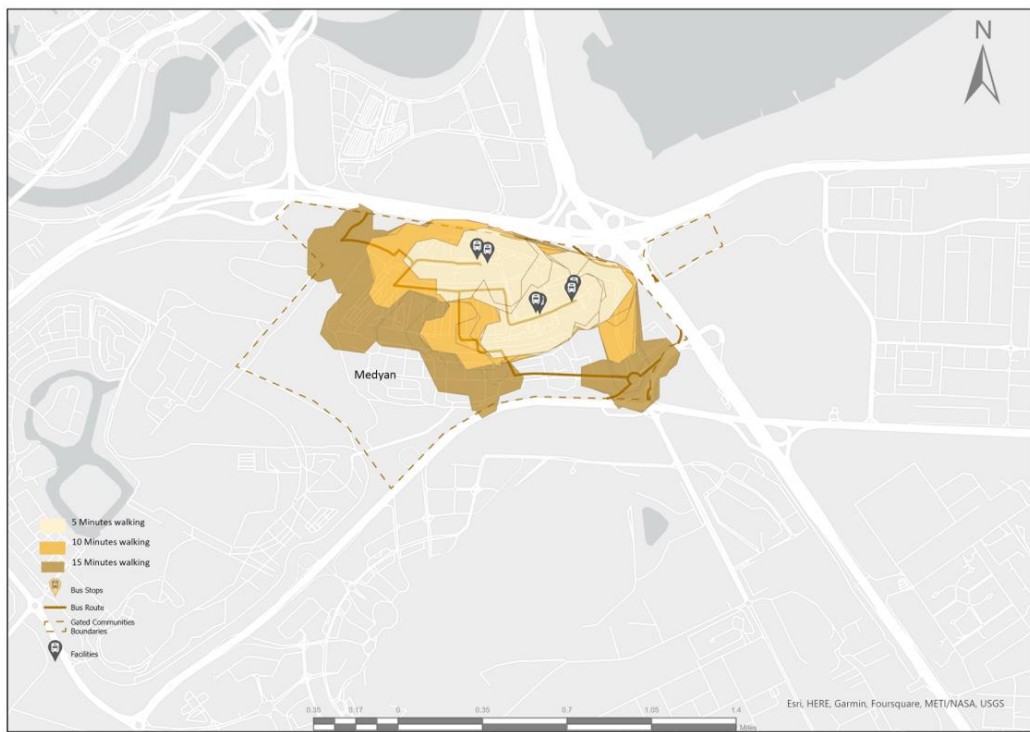

**Figure 20.** The bus stops walkability within 5, 10, and 15 min for Meydan.

The unbalanced availability of the services resulted in the use of combined research methodologies in order to obtain accurate measures of the 15-min walkability, utilizing ArcGIS Pro 3.0, random resident interviews, and site visits. The checklist survey used in this study (Table 3) identified critical information (population density [50,51], mixed land use [52], local facilities and services [44], accessibility [52], safety [53,54], and integra-

tion [50,55]. According to Google Maps' information on the service number, site visits, and interviewing random community residents, the appropriate indicators of those principles are assessed in Table 4.

**Table 4.** The walkability principles and their indicators.

| Principles | Standard Indicators of Walkability | Indicators in Dubai Gated Communities |
|---|---|---|
| Population Density | Medium population density of 4000–5000 population per Km$^2$. | Gated communities vary in population density from 2119 (the Meadows) to 538 (Mudon) population per Km$^2$ (low density) [45]. |
| Mix Land Use | (a) Provision of integrated residential, recreational, and civic uses that are basic to everyday life and diversity of local job opportunities and appropriate workspace. <br> (b) Connection to public transportation. <br> (c) Provision of housing variety. | (a) The integration between the residential and recreational facilities is solid and clear. But all the gated communities need more business, educational, health, and commercial services. <br> (b) 45% of the gated communities are within the bus service network (Figure 17). <br> (c) According to [50], Damac Hills and Dubai Hills are the only two developments that have a wide range of housing typologies (apartments and villas). |
| Local facilities and services | According to Dubai Municipality Planning Standards [56]: <br> (a) Local mosque has a minimum area of 1800 m$^2$ with a service area of 500 m. <br> (b) Retail facilities have a minimum area of 400 with a service area of 400 m. <br> (c) Post shelter has a minimum area of 35 m$^2$ with a service area of 400 m. <br> (d) The local plaza has a minimum area of 1500 m$^2$, a total minimum area, and a neighborhood park of 4000 m$^2$. | (a) The service area of the local mosques ranges from 600 m$^2$ in Arabian Ranches to 1000 m$^2$ in Al Barari. <br> (b) Applicable only in Arabian Ranches. The Meadows, the springs, Damac Hills, and Dubai Hills are considered only one mega shop (in some cases, a mall) to serve the whole community. <br> (c) The service area of the post shelters (when they are available) differs from the walkability standards. <br> (d) Matching the walkability principles in all considerable gated communities. |
| Accessibility | (a) Provision of amenities and public transportation nodes within a walkable distance and accessibility to public transportation. <br> (b) Provision of street furniture, such as soft scapes, shading devices, etc. <br> (c) Accessible buildings and spaces for kids, people with special needs, women, and elderly people. | (a) Only some of the transportation nodes (when available) are within 15 min of walking distance (Figures 18–20). <br> (b) All the neighborhood plazas are provided with benches, and a few of them have shading devices. <br> (c) The retail facilities provide special parking for the elderly and people with disabilities. |
| Safety | (a) Provision of pedestrian sidewalks, crosswalks, and barriers. <br> (b) The minimum sidewalk width, streetlights, availability of traffic calming, and speed limit. <br> (c) Providing an eye on the street concept through having an active street façade and rooms with street view. | (a) The majority of interior roads within the gated communities' boundaries consist of walking lanes and pedestrian paths to cross them. <br> (b) All the standards of the sidewalks are provided on the interior roads. <br> (c) The visibility and having a street view are available through providing windows and balconies, in some cases, to the upper floor level in the cases of villas and apartments. |
| Integration and hierarchy | (a) Connection housing clusters (houses are interconnected with other clusters, neighborhood centers, surrounding areas, and the city). <br> (b) Street capacity intersection density. | (a) Cluster and neighborhood levels are well defined and connected with the different levels of road hierarchy. <br> (b) Streets capacity is strong among the residents without facing daily traffic issues. |

## 4. Discussions, Limitations, and Future Work

According to the ArcGIS network analysis, sites visited, and interviews with random residents, and as expected, the gated communities in this research are not walkable within a

15-min timeframe due to their planning concept that relies on segregation. However, all of these communities provided strong sidewalks with very high safety standards, integration, and other walkability principles. Basic needs services are not provided within a walkable distance, and in some cases, they are not available. Results showed that the Arabian Ranches have the highest resident percentage (80%) who can access most of the services available. In contrast, the Akoya Oxygen, Jumairah Gulf (100%) of their residents cannot access any services within the community boundary (entertainment services are excluded).

It is important to note that a 15-min city assessment should take into account both walkability and cyclability. Future research should consider cyclability in order to create a full assessment of Dubai's sustainable mobility. Moreover, although the gated communities considered in this study provide the hierarchy, integration, capacity, and safety of the walking infrastructure, the ungated communities analysis does not consider that, which, of course, can affect the accessibility time frame. Furthermore, different types of residents for disaggregated users could be considered, such as nursing homes (e.g., for senior citizens, although there are very few). At last, if the facilities' diversity is included, it may promote significant changes in the research outcome.

## 5. Conclusions

This study provides a methodological framework to assess the walkability of the core Dubai area to essential facilities. The future challenges and concerns for the 15-min city could easily be identified by the existing planning threats for the developments based on the walkability ideas, such as Berry's Neighborhood Unit [57]. The walkability concept was conceived for planning new districts and designing new neighborhoods, and moreover, it could be applied to existing ones. In this context, the spatial pattern of the facilities (the ideal one when it is more scattered and less homogeneous between the neighborhoods) could play a pivotal role in solving many accessibility issues at the district and city scale. This study has identified accessibility levels through walking mode for 14 types of local services in gated and ungated communities. Since Dubai is home to different types of urban structures that characterize its developments, study areas were selected to represent the neighborhoods in Dubai. It is necessary to thoroughly explore the Dubai community to accurately evaluate walkability, rather than relying on only a few case studies.

First, the research finds that in a diverse city such as Dubai, it is essential to have different parameters to determine the essential services, which vary from one area to another. This is the obvious result, of course, of the significant difference in population density in Dubai. The concerned urban authorities may need to focus on service capacity to match the urban scale hierarchy to achieve the walkability target by 2040.

Second, in high-density areas, the walkability time varies noticeably from one part of the community to another in the same neighborhood or even for the same service at the same time frame. It is imperative to obtain equal accessibility to essential services to avoid the overpopulation density in some parts of the development.

Third, the accessibility of different services has a major impact on the average population's ability to walk within 15 min. There is a large disparity between the availability of services such as churches and parks, which require residents to rely on cars, and services such as primary schools and grocery stores, which are within a 15-min walking distance. This is especially true in ungated neighborhoods, but it can also be observed in some gated communities, such as Arabian Ranches. While some facilities are inaccessible within 15 min, most are within reach, making it the most walkable gated community in Dubai, according to the research outcomes.

Fourth, even the ungated neighborhoods, where almost three-quarters of their inhabitants could walk within 15 min to access essential services, can only walk within the development boundary. To enhance the walkability in Dubai, the study suggests considering walking infrastructure that connects and integrates multi-development in order to share more facilities and increase the possibility of accessing some mega-scale services, such as malls and universities, by using sustainable transport modes.



Fifth, Bur Dubai and Business Bay are two of the few developments linked together by pedestrian ways, which allow their inhabitants to share many facilities together. For this reason, with special consideration for the walking links between them, the analysis considered them connected, not totally separated.

Finally, in the last decades, the idea of 15-min cities has received much global attention, especially after COP21, among all the sustainability and resilience initiatives. As a result, land use and transportation planning professionals should consider the availability of essential services in planning cities and in solving some spatial, social, economic, and personal issues by affecting the affordability of services and opportunities [58]. The 15 min consider the neglected societal group that cannot use cars because of their physical and financial abilities (children and senior citizens) [59].

**Author Contributions:** All the authors of this manuscript have contributed significantly to the work reported here. Conceptualization, T.A. and M.A.; Methodology, M.A. and R.G.; Software, R.G.; Validation, T.A.; Formal Analysis, M.A.; Investigation, R.G.; Resources, T.A.; Data Curation, T.A.; Writing—Original Draft Preparation, M.A.; Writing—Review and Editing, T.A.; Visualization, M.A. and R.G.; Supervision, T.A.; Project Administration, T.A. and A.E.; Funding Acquisition, T.A. All authors have read and agreed to the published version of the manuscript.

**Funding:** This research was funded by American University of Sharjah through grant number FRG21-M-E77 and the APC was funded by grant number OAPCEN-1410-E00218.

**Informed Consent Statement:** Not applicable.

**Data Availability Statement:** Not applicable.

**Acknowledgments:** Not applicable.

**Conflicts of Interest:** The authors declare no conflict of interest.

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
