# Peer review of "Fifteen-, Ten-, or Five Minute City? Walkability to Services Assessment: Case of Dubai, UAE"

_sustainability, doi:10.3390/su152015176_

Round 1

Reviewer 1 Report

This paper addresses an interesting case study of a location which has considerable accessibility challenges. However, the text, and the accompanying figures and tables, are not particularly clear, making the paper difficult to read and follow. There also seem to be some fundamental errors with the use of accessibility timebands, i.e. instead of considering them separately, 0-5 minutes should be included  within 0-10 minutes, and both within 0-15 minutes. Detailed comments and feedback are provided in the attached, marked-up copy of the paper.

The quality and clarity of the language used must be improved - a thorough proof-reading by someone with a good knowledge of English is recommended.

Author Response

Dear Reviewer 1,

The authors would like to thank you for your thorough review and comments that have enhanced the quality of the manuscript. Please note that the manuscript has been revised to incorporate the comments and suggestions made by you and other  reviewers as much as possible. 

The authors responses to your comments are shown below each comment in Italic text. All the changes in the revised manuscript in response to your comments are highlighted with green color.  Changes in the revised manuscript in response to other reviewers highlighted in yellow, blue, and red. English editing is shown using Track Changes.  

Your comments: (all the changes in the revised manuscript in response to Reviewer 1 comments are highlighted with green color)

  1. Title should be in a consistent font (Line 2).

 Thanks for the comment. Fixed.

  1. Use Sustainability definition by the UN (line 34).

Thanks for the comment. Fixed. And the official UN website was cited for that ( Reference 1)

  1. Using the subscript (line 44).

Thanks for the comment. Fixed.

  1. What does (… declining the climate change chart..) mean? ( line 44,45)

 Thanks for the comment. The sentence was rephrased for clarity as : (… reducing the factors contributing to climate change…)

  1. Incorrect capitalization; ‘Considering’ ( line 47).

 Thanks for the comment. Fixed.

  1. Removing the strike-through from [18] (line 72)

 Thanks for the comment. Fixed.

  1. Check (mood) spelling error (line 82)

 Thanks for the comment. Fixed.

  1. Wrong capitalization ’A’ (line 90)

 Thanks for the comment. Fixed.

  1. Clarify ‘ The actual and future challenge 139 for Dubai neighborhoods to be a 15 minutes city’ (line 140)

Thanks for the comment. Fixed.

The sentence was rephrase for clarity purpose as : ( The current and upcoming challenge for the neighborhoods in Dubai is to ensure that all essential amenities and services are within a 15-minute distance from the residents).

  1. Clarify ‘ neighborhoods have the fastest increase in Dubai’s population’ ( line 141)

Thanks for the comment. Fixed.

The sentence was rephrase for clarity purpose as : ( Dubai's population is growing rapidly and all of the neighborhoods are experiencing the fastest increase in population)

  1. The Population density seems very high ‘762.6 individuals per square meter’. (line 163)

Thanks for the comment. Fixed.

Revised and corrected as ‘ 762.6 individuals per square kilometer’.

  1. Spell out (CAGR). (Line 167)

Thanks for the comment. Fixed.

  1. Avoid the vicious cycle sentence: ‘Dubai is not a very pedestrian-friendly city, 169 which can make automobile traffic quite heavy.’ (line 168)

Thanks for the comment. Fixed.

Corrected as :’ Dubai is a city known for its heavy traffic. It is predominantly an automobile-oriented city.

  1. The research area neighborhoods should be mapped (lines 182-184).

 Thanks for the comment. Fixed.

Figures 5 and 6 have been moved forward as Figures 3 and 4 for clarity and consistency.

  1. Caption should be adjusted ( line 214)

Thanks for the comment. Fixed.

  1. Explain why Dubai boundary is discontinue.

Thanks for the comment. Fixed. Pease refer to the description of Figure 3.

  1. Is the data used in the analysis is confidential and what is the linked between the data and the reference [36].

 Thanks for the comment. Fixed.

Explained as; ‘ using some confidential sources of  data, in addition to the available data online posted by Dubai Municipality website’.

  1. Wrong capitalization (Resoans, Neighbourhoods)

Thanks for the comment. Fixed.

  1. Remove the word (existing).

Thanks for the comment. Fixed.

  1. Wrong use of ‘Geotechnical data’ ( page 11)

Thanks for the comment. Replaced by ‘Georeferenced data’.

  1. ‘Churches’ spelling mistake ( table 1, page 11)

Thanks for the comment. Fixed.

  1. List the paragraph as bullets points( page 12)

Thanks for the comment. Fixed.

  1. Replace the word ‘kids’ with ‘children’, and the word ‘elderlies’ with ‘senior citizens’

Thanks for the comment. Fixed. The change applied to the whole paper.

  1. Revise the numbering in Page 15.

Thanks for the comment. Fixed.

  1. Add the phrase ‘ and one than more service’ to the Table 2 description.(page 15).

Thanks for the comment. Fixed.

  1. Clarify the methodology.

Thanks for the comment. All the methodology section has been revised. New figures have been added for more clarification, and all the section has been rephrased for better understanding.

  1. Rephrase ‘barring’ with ‘segregation’. (page 20).

Thanks for the comment. Fixed.

  1. ‘…more than 15%’ , identify is it per population or area? (page 20).

Thanks for the comment. Fixed.

  1. Move the caption for Figure 16. (page 21)

Thanks for the comment. Fixed.

  1. Table 3 is not clear.

Thanks for the comment. Re-inserting the table 3 with the standard word’s table format.

  1. Re-formatting table 3 headers for legibility.

Thanks for the comment. Fixed.

  1. Explaining why the gated communities can’t be considered as mixed-used neighborhoods. (Table 4)

Thanks for the comment. Fixed.

Not only consisting of some workspaces and residential could be considered as mixed-use. The absence of educational, health, and commercial facilities should be the reason (clarified within the paper text).

  1. Clarify ‘ According to [50], the top real estate websites in Dubai are Damac Hills,…’ ( Table 4).

Thanks for the comment. Fixed.

  1. ‘provide views to the artificial water lakes’ , is irrelevant to safety.(Table 4)

 Thanks for the comment (removed).

  1. It's not surprising that services are not within walkable distances and times.(page 26)

 Thanks for the comment. Fixed.

It have been mentioned within the context is this result is expected. Kindly highlight that one of  the research findings, not all of the gated communities are not walkable (Arabian Ranches illustrates this exception)

  1. Reword: ‘ a 15-minute city assessment should be considered cyclability, besides the walkability’ . (page 26)

Thanks for the comment. Fixed.

  1. Add reference for berry’s neighborhood unit. (page 26)

Done. Reference [58].

  1. Re-word ‘As just 13 case studies have been considered, any claim of systematicity must be included to exhaust the Dubai community and draw a general image of the walkability level’. (page 27)

Thanks for the comment. Fixed.

  1. Re-word ‘few’ with ‘some’. (Page 27, paragraph 3)

Thanks for the comment. Fixed.

  1. Check the spelling for ‘moods’. ( Page 27, paragraph 4)

Thanks for the comment. Fixed.

  1. Meaning of ‘ …take their current essential weight in…’. ( Page 27, paragraph 5)

Thanks for the comment. Fixed.

  1. Remove ‘able to afford.’ (page 28).

Thanks for the comment. Fixed.

Reviewer 2 Report

General evaluation of the article:

·         Title: Suitable

·         Keywords: Suitable

·         Abstract: Suitable.

·         Itemization: Correct

·         Text: clear and objective

·         Article with very interesting current and relevant research. Content very related to the scope of the Journal.

·         Bibliographic research: objective and related to the article theme

·         Conclusions consistent with objectives and research discussed.

Specific Comments:

·         The text aroused several doubts:

a.    The text does not mention whether bicycles travel in dedicated traffic lanes or on the road for motorized vehicles.

b.    The criteria for calculations that relate travel times and the respective distances covered are not presented.

c.    Among the areas studied, which ones were more or less adequate in terms of the concepts applied? If possible, explain these conclusions.

Author Response

Dear Reviewer 2,

The authors would like to thank you for your thorough review and comments that have enhanced the quality of the manuscript. Please note that the manuscript has been revised to incorporate the comments and suggestions made by you and other  reviewers as much as possible. 

The authors responses to your comments are shown below each comment in Italic text. All the changes in the revised manuscript in response to your comments are highlighted with yellow color. Changes in the revised manuscript in response to other reviewers comments are highlighted with green, blue, and red colors. English editing and proofreading is shown with Track Changes. 

  1. The text does not mention whether cyclability travel in dedicated traffic lanes or on the road for motorized vehicles.

 Thanks for the comment. Please note that this study has not considered cyclability, which is a limitation. Future research on the topic should consider cyclability.

  1. The criteria for calculations that relate travel times and the respective distances covered are not presented.

 Thanks for the comment. The calculation criteria and the centroid network analysis map are added to the methodology section. Kindly refer to page 14.

  1. Among the areas studied, which ones were more or less adequate in terms of the concepts applied? If possible, explain these conclusions.

Thanks for the comment.

Please note that out of the 11 gated communities considered, only Arabian Ranches met the walkability standards according to the conclusion. Additionally, this section discusses the current walkability calculations for the un-gated communities of Bur Dubai and Business Bay.

Reviewer 3 Report

The subject is interesting, maybe you can improve the method by optimisation of the parametre, by taking in account other factors like ride&share parkings. This fact can improve the mobility.

I detect some errors:

line 104 - the title of the figure 1 is far away from the picture.

line 108 - double spacing

line 124 - double spacing

line 177 - I think you there is a unwanted space (Un gated -> Ungated)

line 216 - need to corelate title with the figures

figures 8, 11, 16, - need to be careful about spacings

table 2 is on two pages

and you have a big gap after table 4.

Author Response

Dear Reviewer 3,

The authors would like to thank you for your thorough review and comments that have enhanced the quality of the manuscript. Please note that the manuscript has been revised to incorporate the comments and suggestions made by you and other  reviewers as much as possible. 

The authors responses to your comments are shown below each comment in Italic text. All the changes in the revised manuscript in response to your comments are highlighted with blue color. Changes in the revised manuscript in response to other reviewers comments are highlighted with green, yellow, and red colors. English editing and proofreading is shown with Track Changes. 

  1. line 104 - the title of the figure 1 is far away from the picture.

Thanks for the comment. Fixed.

  1. line 108 - double spacing

Thanks for the comment. Fixed.

  1. line 124 - double spacing

Thanks for the comment. Fixed.

  1. line 177 - there is an unwanted space (Un gated -> Ungated)

Thanks for the comment. Fixed.

  1. line 216 - need to correlate title with the figures

Thanks for the comment. Fixed.

  1. figures 8, 11, 16, - need to be careful about spacings

Thanks for the comment. Fixed.

  1. table 2 is on two pages

Thanks for the comment. Fixed.

  1. a big gap after table 4.

Thanks for the comment. Fixed.

Reviewer 4 Report

A review on the manuscript in journal Sustainability entitled „Fifteen-, Ten-, or Five Minutes City? Walkability to Services Assessment: Case of Dubai, UAE“.

The article assesses the current walking situation in the most important parts of Dubai. This study examines 14 essential services, divided into five categories: education, health, social, entertainment and religious.

Broad comments

Research methods have been described at satisfactory level. The conclusions are based on analysis and are adequate.

The formatting of the Tables and Figures in the article needs significant revision.

Specific comments

All Figures must be identified with a number and followed by a brief statement that describes the data provided. Important readings on the Figure can be highlighted in captions. Captions must take place below the figure.

Under each Figure there must be a Figure number and caption, which must be formatted in the same way throughout the article.

Unfortunately, there is a period at the end of the Figure captions (Figures 3, 4, 7, 18, 19, 20) and there is no period (Figures 1, 2, 5, 6, 9, 10, 11, 12, 13,14, 15, 17).

The captions of Figures 1, 8 and 16 are not below the Figure, but in any place within the text.

The caption of Figure 3 is below Figure 4 with the caption of Figure 4.

Table numbers and captions must be above the table. Unfortunately, this has not been taken into account in any of the tables.

Table 3 and Figure 17 cover each other, it does not allow access to information.

Author Response

Dear Reviewer 4,

The authors would like to thank you for your thorough review and comments that have enhanced the quality of the manuscript. Please note that the manuscript has been revised to incorporate the comments and suggestions made by you and other  reviewers as much as possible. 

The authors responses to your comments are shown below each comment in Italic text. All the changes in the revised manuscript in response to your comments are highlighted with red color. Changes in the revised manuscript in response to other reviewers comments are highlighted with green, yellow, and blue colors. English editing and proofreading is shown with Track Changes. 

  1. All Figures must be identified with a number and followed by a brief statement that describes the data provided. Important readings on the Figure can be highlighted in captions. Captions must take place below the figure. Under each Figure there must be a Figure number and caption, which must be formatted in the same way throughout the article.Unfortunately, there is a period at the end of the Figure captions (Figures 3, 4, 7, 18, 19, 20) and there is no period (Figures 1, 2, 5, 6, 9, 10, 11, 12, 13,14, 15, 17).

Thanks for the comment. All the captions of the figures were revised.

  1. The captions of Figures 1, 8, and 16 are not below the Figure but in any place within the text.

Thanks for the comment. All the captions of the figures were revised.

  1. The caption of Figure 3 is below Figure 4 with the caption of Figure 4.

Thanks for the comment. All the captions of the figures were revised.

  1. Table numbers and captions must be above the table. Unfortunately, this has not been taken into account in any of the tables.

 Thanks for the comment. All the captions of the tables were revised.

  1. Table 3 and Figure 17 cover each other, it does not allow access to information.

Thanks for the comment. Fixed.

Round 2

Reviewer 1 Report

Thank you for updating the paper is response to earlier feedback - the latest revision is a considerable improvement. Some residual comments are provided in the attached, marked-up copy of the paper, and the 'Available services and the population' part of section 3.1, in particular, needs work to improve its clarity.

The quality of the English is considerably better than in the previous draft - some specific comments have been made in the attached, marked-up copy of the paper, and there is still room for general improvement. 

Author Response

Dear Reviewer 1,
The authors would like to thank you for your thorough review and comments that have considerably enhanced the quality of the manuscript. Please note that the manuscript has been revised to incorporate your comments and suggestions as much as possible. Please note that all the changes in the revised manuscript in response to your comments are highlighted with green color in the revised manuscript and our responses to your comments are shown below:

  1. Capitalize ’d’ in ‘during’ (page 4, line 24)

      Done.

  1. Use the term ‘technology’ instead of ‘ technical’ (page 4, line 28).

     Done.

  1. The caption of Figure 4 (page 6), It still doesn't show Bur Dubai explicitly.

      Done. The figure caption has been changed.

  1. Check (layers) spelling (page 8, line 1).

     Done.

  1. Remove the strikethrough on (page 8, line 10).

     Done.

  1. Wrong use for the single bullets (page 8, lines 18-20)

     Done.

  1. Why services such as libraries and social consultation centers were excluded? (page 8, line 28).

     Done. An explanation for the reason has been provided.

  1. Why are supermarkets and open markets considered inessential? (page 8, line 29).

     Done. Rephrased for clarity.

  1. ‘The results seem to indicate that some police stations are within 15 minutes’ (page 13, line 4).

     Done. The whole paragraph was revised for clarity.

  1. (71.50%) should be (75.75%) (page 13, line 11).

    Done.

  1. The section (3.1) remains somewhat unclear - as mentioned previously, surely the work should be looking at 0-5, 0-10 and 0-15-minute accessibility rather than 0-5, 5-10 and 10-15?

Kindly note that the analysis has been updated to include intervals of 0-5, 0-10, and 0-15 minutes instead of 0-5, 5-10, and 10-15 from the last revision. Figure 15 was wrongly labeled. Please note that the figure has been corrected accordingly.

  1. There seems to be a Figure missing, and/or the positions of the Figure captions are inconsistent (Page 20).

    Done. Figures 17 and 18 are inserted correctly in their place.

  1. Put space ( page 23 , line 28)

    Done.

  1. The concluding points should either be given separate paragraphs or set out as bullets/a numbered list ( page 23).

    Done.

Reviewer 4 Report

A review on the manuscript in journal Sustainability entitled „Fifteen-, Ten-, or Five Minutes City? Walkability to Services Assessment: Case of Dubai, UAE“.

The article assesses the current walking situation in the most important parts of Dubai. This study examines 14 essential services, divided into five categories: education, health, social, entertainment and religious.

Broad comments

The article has been significantly supplemented and corrected.

Research methods have been described at satisfactory level. The conclusions are based on analysis and are adequate.

Specific comments

Unfortunately, there is no period at the end of the Figure titles (Figures 2, 5, 8, 11, 12, 13, 15, 16, 17 7, 18, 19, 20).

There is no period at the end of the Table titles (Table 1, 2, 3, 4).

All figures and tables should be referenced in the text, unfortunately there is no reference to Figure 8 in the text.

It is not good style to use the phrase "Shows" in figure captions. Figures are essentially graphical representations of results/situation.

Some references to sources could be presented more briefly, instead of [25,26,27,28] could be [25-28] and instead of [7,16,38,39, 40,41,42,43,44] could be [7,16, 38-44].

Author Response

Dear Reviewer 4,
The authors would like to thank you for your thorough review and comments that have considerably enhanced the quality of the manuscript. Please note that the manuscript has been revised to incorporate your comments and suggestions as much as possible. Please note that all the changes in the revised manuscript in response to your comments are highlighted with red color in the revised manuscript and our responses to your comments are shown below:

  1. Unfortunately, there is no period at the end of the Figure titles (Figures 2, 5, 8, 11, 12, 13, 15, 16, 17, 7, 18, 19, 20).

     Done.

  1. There is no period at the end of the Table titles (Table 1, 2, 3, 4).

     Done.

  1. All figures and tables should be referenced in the text. Unfortunately, there is no reference to Figure 8 in the text.

     Done.

  1. It is not a good style to use the phrase "Shows" in figure captions. Figures are essentially graphical representations of results/situations.

       Done. The Phrase “shows” is removed from all the figure's captions.

      The captions are rephrased to indicate or represent a result/ situation.

  1. Some references to sources could be presented more briefly, instead of [25,26,27,28] could be [25-28], and instead of [7,16,38,39, 40,41,42,43,44] could be [7,16, 38-44].

     Done.